# Molecular Detection of Fluoroquinolone Resistance among Multidrug-, Extensively Drug-, and Pan-Drug-Resistant *Campylobacter* Species in Egypt

**DOI:** 10.3390/antibiotics10111342

**Published:** 2021-11-03

**Authors:** Ahmed M. Ammar, Marwa I. Abd El-Hamid, Rania M. S. El-Malt, Doaa S. Azab, Sarah Albogami, Mohammad M. Al-Sanea, Wafaa E. Soliman, Mohammed M. Ghoneim, Mahmoud M. Bendary

**Affiliations:** 1Department of Microbiology, Faculty of Veterinary Medicine, Zagazig University, Zagazig 44519, Egypt; prof.ahmedammer_2000@yahoo.com (A.M.A.); mero_micro2006@yahoo.com (M.I.A.E.-H.); 2Animal Health Research Institute-Agriculture Research Center, Zagazig University, Zagazig 44516, Egypt; raniaelmalt@yahoo.com; 3Zagazig Veterinary Hospital, Zagazig University, Zagazig 44516, Egypt; gost_66@yahoo.com; 4Department of Biotechnology, College of Science, Taif University, Taif 11099, Saudi Arabia; dr.sarah@tu.edu.sa; 5Pharmaceutical Chemistry Department, College of Pharmacy, Jouf University, Sakaka 72341, Saudi Arabia; mmalsanea@ju.edu.sa; 6Department of Biomedical Sciences, College of Clinical Pharmacy, King Faisal University, Al Ahsa 31982, Saudi Arabia; wafaaezz2006@yahoo.com; 7Microbiology and Immunology Department, Faculty of Pharmacy, Delta University for Science and Technology, Gamasa 35712, Egypt; 8Department of Pharmacy Practice, College of Pharmacy, Al Maarefa University, Ad Diriyah 13713, Saudi Arabia; mghoneim@mcst.edu.sa; 9Department of Microbiology and Immunology, Faculty of Pharmacy, Port Said University, Port Said 42511, Egypt

**Keywords:** *Campylobacter* species, PDR, FQ resistant, *gyrA*, PCR-RFLP

## Abstract

In recent times, resistant foodborne pathogens, especially of the *Campylobacter* species, have created several global crises. These crises have been compounded due to the evolution of multidrug-resistant (MDR) bacterial pathogens and the emergence of extensively drug-resistant (XDR) and pan-drug-resistant (PDR) strains. Therefore, this study aimed to investigate the development of resistance and the existence of both XDR and PDR among *Campylobacter* isolates. Moreover, we explored the use of the polymerase chain reaction–restriction fragment length polymorphism (PCR-RFLP) technique for the detection of fluoroquinolone (FQ)-resistant *Campylobacter* isolates. A total of 120 *Campylobacter* isolates were identified depending on both phenotypic and genotypic methods. Of note, cefoxitin and imipenem were the most effective drugs against the investigated *Campylobacter* isolates. Interestingly, the majority of our isolates (75%) were MDR. Unfortunately, both XDR and PDR isolates were detected in our study with prevalence rates of 20.8% and 4.2%, respectively. All FQ-resistant isolates with ciprofloxacin minimum inhibitory concentrations ≥4 µg/mL were confirmed by the genetic detection of *gyrA* chromosomal mutation via substitution of threonine at position 86 to isoleucine (Thr-86-to-Ile) using the PCR-RFLP technique. Herein, PCR-RFLP was a more practical and less expensive method used for the detection of FQ resistant isolates. In conclusion, we introduced a fast genetic method for the identification of FQ-resistant isolates to avoid treatment failure through the proper description of antimicrobials.

## 1. Introduction

Campylobacteriosis, caused by *Campylobacter* species (spp.), is a worldwide foodborne bacterial disease with zoonotic importance. Most of the human campylobacteriosis cases are caused by *Campylobacter jejuni* (*C. jejuni*) and the closely related *Campylobacter coli* (*C. coli*). Since 2005, campylobacteriosis is the most commonly reported gastrointestinal infection in humans in the European Union (EU) [1,2,3]. In developing countries, this disease is hyperendemic, especially in young children and infants [4,5]. The main source of human *Campylobacter* infection is the consumption of raw or uncooked meat, contaminated water and unpasteurized milk. Chicken is considered the main reservoir of *Campylobacter* infections because these bacteria are commensal in their intestinal tracts due to their high body temperature [4,6,7].

Furthermore, cross-contamination with *Campylobacter* spp. is common during food processing and storing. The consumer’s bad hygienic practices, such as cleaning raw chicken with water and using contaminated kitchen utensils, are considered the main causes of the wide spreading of campylobacteriosis. Additionally, poor hygiene especially during the defrosting and storing of chickens may increase the possibility of cross-contamination by this pathogen [2,3,5].

Campylobacteriosis is a self-limiting disease with symptoms such as fever, nausea, vomiting, abdominal pains and watery or bloody diarrhea, but postinfection complications can occur such as Guillain–Barré syndrome, Miller Fisher syndrome and reactive arthritis in the case of immunocompromised patients [8,9]. In the majority of campylobacteriosis cases, antimicrobial treatment is not indicated, but in severe infections and immunocompromised persons, antimicrobials become of high importance [10]. Therefore, treatment failure in the severe cases may lead to death. Macrolides such as erythromycin and, fluoroquinolones (FQs) such as ciprofloxacin, are the drugs of choice in the treatment of infected cases [11]. Increasingly, a higher incidence of resistant *Campylobacter* spp. is detected worldwide due to the uncontrolled usage of antimicrobials in livestock breeding and through horizontal transmission in the poultry industry [4,12]. *Campylobacter* spp. have been described to be resistant to several antimicrobial classes including FQs, beta-lactams, tetracyclines and aminoglycosides, which leads to an increase in the number of infections with MDR *Campylobacter* strains [13].

Fluoroquinolones depend on the inhibition of the DNA gyrase, which is responsible for DNA repair, recombination, transcription and replication, and topoisomerase IV enzyme, leading to the generation of double breaks of DNA and cell death. On the other hand, two mechanisms explain the resistance to FQs; inactivation of the FQ targets and efflux of the drug. The inactivation process is mainly due to chromosomal point mutations in the quinolone-resistance-determining region (QRDR) of the *gyrA* gene mostly by substitution of threonine at position 86 to isoleucine (Thr-86-to-Ile); this alteration is always associated with high MIC values for FQs. Additionally, the resistance nodulation cell division superfamily efflux pump has been reported to play a role in the resistance to FQs [12,14,15]. The World Health Organization (WHO) announced in 2017 that fluoroquinolone-resistant *Campylobacter* spp. have been increasing all over the world, therefore alternative effective antibiotics must be found [16].

Nowadays, molecular techniques are of great importance in detecting the mutations in specific genes, which are responsible for the antimicrobial resistance. These techniques include denaturing gradient gel electrophoresis, DNA sequencing of the target gene, whole-genome sequencing, non-radioisotopic single-strand confirmatory polymorphism and a fluorogenic PCR assay as direct methods used for detection of the mutation, but they cannot be used as routine protocols for diagnosis as they are expensive and take a long time [12,15]. Therefore, alternative techniques such as PCR-based restriction fragment length polymorphism (PCR-RFLP) are more important as they are rapid, more practical and less expensive when they are used for the detection of mutations by the digestion of the PCR products using restriction enzyme and a specific restriction-site-modified primer. This technique was used as a diagnostic method for FQ-resistant *Campylobacter* spp. by determining the target mutations in QRDR of the *gyrA* gene [17]. There is a little information about the FQ resistance mechanism among *Campylobacter* species in Egypt [12,14]. Therefore, the current study aimed to get information about the occurrence and the resistance rates of *Campylobacter* isolates in Sharkia Governorate, Egypt. Additionally, we planned to provide the clinicians with a rapid detection method for FQ-resistant *Campylobacter* species using the PCR-RFLP technique, which will aid the physicians to make fast treatment decisions.

## 2. Results

### 2.1. Prevalence of Campylobacter Species in Different Samples at Sharkia Governorate, Egypt

According to the phenotypic identification, a total of 120 *Campylobacter* isolates (57.1%) were obtained from 210 different samples at Sharkia Governorate, Egypt (Table 1). *Campylobacter* spp. were prevalent among human (85.7%) and chicken (51.4%) samples. Among chicken samples, *Campylobacter* isolates were more prevalent among cloacal swabs and liver (88.6% each), while chicken franks and luncheon meat were *Campylobacter* negative (Table 1).

Regarding the species level, *C. jejuni* was the predominant species (45.7%), followed by *C. coli* (11.4%). The highest isolation rates of *C. jejuni* isolates were observed in human stool swabs (71.4%), followed by chicken cloacal swabs and breast muscle samples (68.6% each). Meanwhile, the highest isolation rates of *C. coli* isolates were detected in the chicken liver (22.9%), followed by human stool swabs (14.3%) (Table 1). Furthermore, there were statistically significant differences (*p* < 0.001) in the prevalence of *Campylobacter* spp., *C. jejuni* and *C. coli* among different sample types.

### 2.2. Antimicrobial Susceptibility Testing of Campylobacter Isolates

#### 2.2.1. Antimicrobial Susceptibility Profiles of *Campylobacter* Species from Various Sources

Analysis of the antimicrobial susceptibility of the recovered 120 *Campylobacter* isolates against the examined 24 antimicrobials showed that all the tested isolates were resistant to amoxycillin, ampicillin, cephalothin and erythromycin. Moreover, high resistance rates were detected against trimethoprim-sulfamethoxazole (98.3%), followed by nalidixic acid (97.5%), clarithromycin (96.7%) and azithromycin and clindamycin (95% each). On the other hand, the lowest resistance rates were observed against amikacin (39.2%), imipenem (40.8%) and cefoxitin (45.8%) (Table 2).

Regarding the species level, *C. jejuni* isolates were 100% resistant to nalidixic acid, trimethoprim-sulfamethoxazole and clarithromycin, while *C. coli* isolates were 100% resistant to clindamycin. Additionally, our results showed higher resistance rates of *C. jejuni* than of *C. coli* isolates for the tested antimicrobials except for cefoxitin, cefoperazone, imipenem, doxycycline, azithromycin, tobramycin, amikacin, colistin and clindamycin (Figure 1). There were statistically significant differences in the resistance prevalence among *C. jejuni* and *C. coli* isolates against sulbactam-ampicillin, amoxycillin-clavulanic acid, cefoxitin, ciprofloxacin, trimethoprim-sulfamethoxazole and chloramphenicol (*p* = 0.01, 0.023, 0.011, 0.016, 0.039 and 0.018, respectively). Additionally, there were higher significant differences in the resistance prevalence among *C. jejuni* and *C. coli* isolates for cefepime, nalidixic acid, clarithromycin and gentamicin (*p* = 0.005, 0.007, 0.001 and 0.003, respectively). Meanwhile, there were no statistically significant differences in the resistance profiles among *C. jejuni* and *C. coli* isolates for the other tested antimicrobials (*p* > 0.05) (Figure 1).

According to the isolates’ sources, higher resistance rates were observed among human *Campylobacter* isolates than the chicken ones for the investigated antimicrobials except for sulbactam-ampicillin, amoxycillin-clavulanic acid, imipenem, aztreonam, trimethoprim-sulfamethoxazole, clarithromycin and amikacin (Figure 2). There were statistically significant differences in the resistance prevalence among human and chicken *Campylobacter* isolates against trimethoprim-sulfamethoxazole, linezolid and colistin (*p* = 0.041, 0.038 and 0.021, respectively). Moreover, there were higher significant differences in the resistance prevalence among human and chicken *Campylobacter* isolates against cefepime, aztreonam (*p* = 0.009 and 0.008, respectively) and cefoxitin (*p* < 0.001). Meanwhile, there were no statistically significant differences in the resistance profiles among human and chicken *Campylobacter* isolates for the other tested antimicrobials (*p* > 0.05) (Figure 2).

Interestingly, our results showed that among human samples, *C. jejuni* isolates were resistant to 9 (16%) and 10 (84%) antimicrobial classes, while *C. coli* isolates were resistant to 7 (40%), 8 (20%), 9 (20%) and 10 (20%) antimicrobial classes. Among chicken isolates, *C. jejuni* isolates were resistant to 7 (12.7%), 8 (19.7%), 9 (31%) and 10 (36.6%) antimicrobial classes, while *C. coli* isolates were resistant to 6 (10.5%), 8 and 9 (15.8% each) and 10 (57.9%) antimicrobial classes. In total, the resistance rates to seven, eight and nine antimicrobial classes were higher in chicken isolates (10%, 18.9% and 27.8%, respectively) than in the human ones (6.7%, 3.3% and 16.7%, respectively). Meanwhile, the resistance to 10 antimicrobial classes was higher in human isolates (73.3%) than in the chicken ones (41.1%) (Figure 3).

Of note, it was found that 90 *Campylobacter* isolates (75%) were MDR, while 25 isolates (20.8%) were recognized as XDR; 6 (20%) and 19 (21.1%) were obtained from human and chicken samples, respectively. Finally, five *Campylobacter* isolates (4.2%) were classified as PDR; four (13.3%) and one (1.1%) were obtained from human and chicken origins, respectively. Determining the MAR indices showed that all tested human isolates had an index of 0.63 or greater, while chicken isolates had an index of 0.50 or greater, which indicate high-risk sources of contamination, where antimicrobial agents are usually utilized (Table 3). Out of 30 *Campylobacter* isolates that had MAR indices greater than 0.9 (resistance to 22 or more antimicrobials), 20 (22.2%) and 10 (33.3%) were isolated from chickens and human samples, respectively (Table 3). There were statistically significant differences in the resistance patterns to 24 antimicrobials and 10 antimicrobial classes among *Campylobacter* isolates from chicken and human origins (*p* = 0.014 and 0.003, respectively) (Figure 3A). Moreover, there were statistically significant differences in the resistance patterns to 14 antimicrobials and 6 antimicrobial classes among *C. jejuni* and *C. coli* isolates (*p* = 0.039 and 0.039, respectively) (Figure 3B).

#### 2.2.2. The Minimum Inhibitory Concentrations of Ciprofloxacin against *Campylobacter* Isolates

Thirty-eight ciprofloxacin-resistant *Campylobacter* isolates that were resistant to 21 or more antimicrobials were tested against ciprofloxacin antibiotic by the broth microdilution method for the determination of its minimum inhibitory concentrations (MICs). Those isolates were recovered from human (13) and chicken (25) sources. The 25 chicken isolates were obtained from the chicken liver (11) and cloacal swabs and breast meat (7 each) samples. Interestingly, all the 38 tested isolates were 100% resistant to ciprofloxacin (MIC ≥ 4 µg/mL) (Table 4) and these results were 100% correlated with those of the disc diffusion method. Additionally, the minimum bactericidal concentration (MBC) values of ciprofloxacin ranged from 8 to ≥256 µg/mL.

### 2.3. Molecular Grouping of Campylobacter Isolates from Different Sources

All the 38 screened *Campylobacter* isolates (100%) were identified as genus *Campylobacter* (Figure 4). Moreover, 29 isolates (76.3%) were positive for *mapA* gene and confirmed to be *C. jejuni* (Figure 5A), while the remaining 9 isolates (23.7%) were positive for the *ceuE* gene and confirmed to be *C. coli* (Figure 5B). These results were 100% correlated with those of the conventional identification methods. Of the 29 *C. jejuni* isolates, 11 (37.9%) were obtained from human and 18 (62.1%) from chicken sources. Moreover, nine *C. coli* isolates were obtained from seven chicken (77.8%) and two human (22.2%) samples. There were statistically significant differences in the prevalence of *C. jejuni* and *C. coli* isolates among human stool swabs and chicken breast meat samples (*p* = 0.001 and 0.029, respectively). Meanwhile, there were no statistically significant differences in the prevalence of *C. jejuni* and *C. coli* isolates among chicken liver and cloacal swabs samples (*p* = 0.395 and 0.286, respectively) (Figure 6).

### 2.4. Determination of Fluoroquinolone Resistance by PCR-RFLP Technique

All the 38 tested *Campylobacter* isolates had the same RFLP fragments (179 bp) (Figure 7), which suggested having a chromosomal point mutation in the QRDR of the *gyrA* gene by substitution of threonine amino acid at position 86 to isoleucine (Thr-86-to-Ile). Additionally, there was a 100% correlation between the ciprofloxacin MIC values and the PCR-RFLP results.

## 3. Discussion

It was announced that several worldwide crises were developed due to the wide spreading of resistant fungi [18] and bacteria such as MRSA, VRSA and *Klebsiella* spp., in addition to zoonotic foodborne pathogens including *Campylobacter* spp., *Salmonella enteritidis* and *Salmonella typhimurium* [10,19,20,21,22,23,24,25,26]. The increasing antimicrobial resistance of *Campylobacter* spp., especially to FQs, macrolides and tetracyclines, is of great importance to human health worldwide. Our results revealed a high prevalence of *Campylobacter* spp. (57.1%) in samples recovered from chicken and human origins at Sharkia Governorate, Egypt. This is partially similar to a previous study carried out in Poland (54.4%) [27], but the levels were higher than those obtained in previous studies carried out in Egypt; 32.8% [10], 27.3% [28] and 7.6% [29]. Herein, *Campylobacter* spp. were more prevalent among human samples, followed by chicken ones, which was in contrast with previous studies conducted in Egypt, where *Campylobacter* spp. were more prevalent among chicken samples, followed by human ones [29,30]. Among our chicken samples, *Campylobacter* spp. were more prevalent among cloacal swabs and liver (88.6% each), which is higher than prevalences reported in a previous study carried out in Egypt (54.3% and 34.1%, respectively) [10]. In the current study, chicken franks and luncheon meats were *Campylobacter* negative (0% each), which is in complete agreement with a previous study conducted in Egypt [31]. Of note, the most common *Campylobacter* spp. was *C. jejuni* (45.7%), which is in complete agreement with the results of other studies carried out in Tunisia (68.9%) [32] and South Korea (77.6%) [33]. Generally, the variations in the prevalence of *Campylobacter* spp. among various studies could be due to the type of the tested samples, hygienic measures, isolation and identification methods, environmental conditions and the geographical location [34].

Unsurprisingly, there is a variation in the antimicrobial resistance among and within different countries, which strongly correlated with the type of prescribed drugs alongside the variation in guidelines for the use of antimicrobial drugs. In this context, the high levels of ciprofloxacin and doxycycline resistance rates detected among our tested isolates (76.7% and 84.2%, respectively) were lower than those detected in a previous study carried out in Tunisia (99.2% and 100%, respectively) [35]. Alarmingly, there has been a global warning concerning the evolution of MDR strains; however, concrete steps are being taken against the spread of both XDR and PDR strains. Herein, 75%, 20.8% and 4.2% of the tested *Campylobacter* isolates were recognized as MDR, XDR and PDR, respectively. This is consistent with the results of a previous study conducted in Egypt, where the above-mentioned resistance criteria were observed among 28.5%, 69% and 2.5% of the tested isolates, respectively [31]. In the current study, all *Campylobacter* isolates had MAR indices of 0.5 or greater. These results were higher than those recorded in a previous study conducted in South Africa, where MAR indices of the tested isolates were 0.2 or lower [36]. The high resistance rates of *Campylobacter* isolates in developing countries could be due to the uncontrolled usage of antimicrobials in veterinary medicine as growth promoters and in human and animal treatments without any prescription. Additionally, the high resistance rates observed in the present study to erythromycin, ciprofloxacin and doxycycline are alarming as these antibiotics are the drugs of choice used for the treatment of human campylobacteriosis, which causes a fundamental problem, where antibiotic treatment become limited. Therefore, antimicrobial usage must be controlled in animals and humans. Additionally, there is an urgent need for the wide application of alternative drugs from medicinal plants [37,38] or drug repurposing [39].

One of the best drugs available for treating human campylobacteriosis is FQ antibiotics. Many physicians describe FQs as a first-line therapy [11,12,15,17]. Unfortunately, there is widespread FQ resistance due to mutations in the gyrase gene [15]. Clinically, it is very important to find a rapid genetic method for the detection of gyrase gene mutations instead of sequencing methods. In the present study, the PCR-RFLP results for all examined isolates revealed that they all had the same fragment pattern (179 bp), which confirmed the resistance to ciprofloxacin and suggested the presence of a chromosomal point mutation in the QRDR of the *gyrA* gene by substitution of threonine amino acid at position 86 to isoleucine (Thr-86-to-Ile). Accordingly, there was a strong direct correlation between the MIC values of ciprofloxacin and the PCR-RFLP results. This result was in complete agreement with a previous study conducted in Brazil [15], which can help in controlling the FQ resistance and allow the proper usage of antibiotics.

## 4. Materials and Methods

### 4.1. Ethical Statement

The sole aim of the specimen collection in this study was to care for patients and to perform antimicrobial susceptibility testing for proper diagnosis and treatment. Therefore, it was not necessary to take the ethical approval, but prior to starting the study, the participants provided informed consent.

### 4.2. Sample Collection

This study was carried out between March 2017 and September 2018 (18 months) at Zagazig city, Sharkia Governorate, Egypt. A total of 210 samples were collected from chicken (*n* = 175) and human (*n* = 35) sources. Ten samples from chickens of 6 weeks of age were collected per month from multiple retail outlets (*n* = 16, 11 samples from each outlet) including cloacal swabs, breast muscles, liver, chicken franks and chicken luncheon meats (*n* = 35 each), while human stool samples were obtained as swabs from diseased children with diarrhea from private laboratories at Zagazig city. Cloacal and stool swabs were transferred directly into a sterile tube containing 9 mL of Bolton broth with Bolton broth selective supplement (Oxoid, UK) with 20 mm space lift in the tube to achieve microaerophilic conditions, while other samples were transferred into ice boxes for laboratory isolation and identification of *Campylobacter* spp.

### 4.3. Isolation and Identification of Campylobacter Species

Isolation of *Campylobacter* spp. was achieved according to the International Standards Organization (ISO) guidelines [40]. Briefly, the enrichment broth containing samples was incubated for 48 h at 42 °C in darkness under microaerophilic conditions (5% O_2_, 10% CO_2_ and 85% N_2_) using CampyGen sachets (Oxoid, Cheshire, UK) and anaerobic jar (Oxoid, Cheshire, UK). After that, 10 µL of the enrichment broth was streaked onto the surface of the selective modified charcoal cefoperazone deoxycholate agar (mCCDA) plates with CCDA selective supplement (Oxoid, Cheshire, UK), and the plates were incubated for 48 h at 42 °C in darkness under microaerophilic conditions. For more purification, suspicious colonies were cultivated onto blood agar (Oxoid, Cheshire, UK) supplemented with 5% sterile defibrinated sheep blood, and the plates were incubated for 48 h under microaerophilic conditions. The *Campylobacter* isolates were presumptively confirmed via cultural characteristics on mCCDA, Gram’s staining, motility, some biochemical tests such as catalase, oxidase and indoxyl acetate and sodium hippurate hydrolysis tests and finally susceptibility to nalidixic and cephalothin.

### 4.4. Antimicrobial Susceptibility Testing

#### 4.4.1. Disc Diffusion Method

Kirby–Bauer disc diffusion method [41] was used to determine the susceptibility of all *Campylobacter* isolates to 24 antimicrobials belonging to 10 different antimicrobial classes that were regularly used in human and veterinary medicine in Egypt using the following antimicrobial discs (Oxoid, Cheshire, UK); amoxycillin (AX, 25 µg), ampicillin (AM, 10 µg), sulbactam-ampicillin (SAM, 10 + 10 µg), amoxycillin-clavulanic acid (AMC, 20 + 10 µg), cephalothin (KF, 30 µg), cefoxitin (FOX, 30 µg), cefoperazone (CEP, 75 µg), cefepime (FEP, 30 µg), imipenem (IMP, 10 µg), aztreonam (ATM, 30 µg), nalidixic acid (NA, 30 µg), ciprofloxacin (CIP, 5 µg), trimethoprim-sulfamethoxazole (SXT, 23.75 + 1.25µg), doxycycline (DO, 30 µg), erythromycin (E, 15 µg), azithromycin (AZM, 30 µg), clarithromycin (CLR, 15 µg), tobramycin (TOB, 10 µg), gentamicin (CN, 10 µg), amikacin (AK, 30 µg), linezolid (LNZ, 30 µg), chloramphenicol (C, 30 µg), colistin (CT, 10 µg) and clindamycin (DA, 2 µg). Few (3–10) colonies were added to a tube containing 5 mL of sterile physiological saline (0.9%) to make the bacterial suspension, which was compared with 0.5 McFarland standard solution. After that, the prepared suspension was streaked on the surface of Mueller-Hinton agar (Oxoid, Cheshire, UK) supplemented with 5% sterile defibrinated sheep blood and then the discs were placed into the plates, which were inverted and incubated at 42 °C for 24–48 h in darkness under microaerophilic conditions. The degree of susceptibility was determined by measuring the visible inhibition zones, and the results were interpreted according to the breakpoints of the Clinical Laboratory Standard Institute (CLSI) to categorize the antimicrobial agents into resistant, intermediate or susceptible [42,43].

The MDR *Campylobacter* isolates were defined as isolates that showed resistance to one or more antimicrobials in at least three different classes, while the isolates resistant to one agent from all antimicrobial categories with the exception of one or two categories were termed as extensively drug resistant (XDR). Meanwhile, the isolates resistant to all agents in all antimicrobial categories were described as pan drug resistant (PDR) [44]. Finally, the multiple antibiotic resistance (MAR) indices of the isolates were determined by the following equation: number of antibiotics to which the isolates were resistant/the total number of antibiotics used [45].

#### 4.4.2. Broth Microdilution Method for Determining Ciprofloxacin Minimal Inhibitory Concentrations

The double-fold broth microdilution method was used to determine the MIC values of ciprofloxacin (EPPI, Cairo, Egypt) using Mueller-Hinton broth (Oxoid, Cheshire, UK) as per the CLSI guidelines [42]. Double-fold serial dilution of ciprofloxacin was done in the 96-well microtiter plates (Techno Plastic Products, Trasadingen, Switzerland) with the initial concentration of 256 μg/mL. The microtiter plates were incubated at 42 °C in darkness under microaerobic conditions for 48 h. The MIC of ciprofloxacin was defined as the lowest concentration that prevented the visible growth of *Campylobacter* isolates after 48 h incubation [46], and the MBC of ciprofloxacin was defined as the lowest concentration that killed 99.9% of the bacterial population after 48 h incubation [47]. Positive control was included through preparing the wells with bacterial inoculum without ciprofloxacin, while negative control was done by preparing wells containing ciprofloxacin without bacterial inoculum. Results of the broth microdilution method were demonstrated according to the CLSI breakpoints for ciprofloxacin resistance (MIC ≥ 4 µg/mL) [42]. Additionally, MIC_50_ and MIC_90_ of ciprofloxacin were identified as the lowest concentrations, which inhibit 50% and 90% of the examined bacterial isolates, respectively [48].

### 4.5. Conventional PCR and PCR-RFLP Assays

Total DNA extraction was carried out using QIAamp DNA Mini Kit (Qiagen, Chatsworth, CA, USA) according to the manufacturer’s instructions. Three pairs of oligonucleotide primers (Metabion, Bayern, Germany) targeting *23S rRNA, mapA* and *ceuE* genes were used in conventional uniplex PCR assays for verification of the genera *Campylobacter, C. jejuni* and *C. coli*, respectively. Moreover, PCR-RFLP was used to determine FQ-resistant *Campylobacter* isolates by detecting the mutation in QRDR of *gyrA* gene using FastDigest *Rsa*I restriction enzyme (Thermo Fisher, Bremen, Germany) [15,17]. All PCR reactions were done in triplicate, using Emerald Amp GT PCR Master Mix (Takara, Kyoto, Japan). according to the manufacturer’s instructions. Sequences of oligonucleotide primers used in all PCR assays are shown in Table 5 [17,49,50]. Agarose gel electrophoresis for visualization of PCR products were done as previously described [51]. Reference strains of *C. jejuni* (NCTC11322) and *C. coli* (NCTC11366) were used as positive controls, and PCR-grade water (no template DNA) was used as a negative control in PCR assays.

### 4.6. Statistical Analysis

The data were analyzed using SPSS version 26 (IBM Corp, Armonk, NY, USA). The chi-square test was used to study the variations in the prevalence of *Campylobacter* spp. from different origins and to assess the differences in the antimicrobial resistance patterns of the recovered isolates from various sources and among *C. jejuni* and *C. coli*. The *p* values were considered statistically significant if they were less than 0.05.

## 5. Conclusions

This study revealed a high prevalence of MDR *Campylobacter* spp. in addition to the existence of both XDR and PDR strains among human and chicken isolates in Egypt. Therefore, there is an urgent need to control the antimicrobial usage in animal production and to improve the hygienic control strategies during slaughtering and carcass processing to reduce the occurrence of resistant *Campylobacter* strains. Interestingly, PCR-RFLP technique was found to be helpful in the detection of FQ-resistant *Campylobacter* isolates, which offer useful insights into the molecular mechanism involved in the FQ resistance in *Campylobacter* spp., which will help public health specialists in the management of *Campylobacter* infections. Furthermore, enhanced research efforts are required to illustrate the mechanisms of the transmission and persistence of FQ-resistant *Campylobacter* spp. alongside different hosts. Complementary to that, it is essential to provide a solid guideline for the administration of FQ antimicrobials especially in the veterinary field, which will decrease the wide spreading of human FQ-resistant *Campylobacter* isolates. A major limitation for our approach is the need for DNA sequencing to confirm the results of the mutation in QRDR of gyrA gene.

## Figures and Tables

**Figure 1 antibiotics-10-01342-f001:**
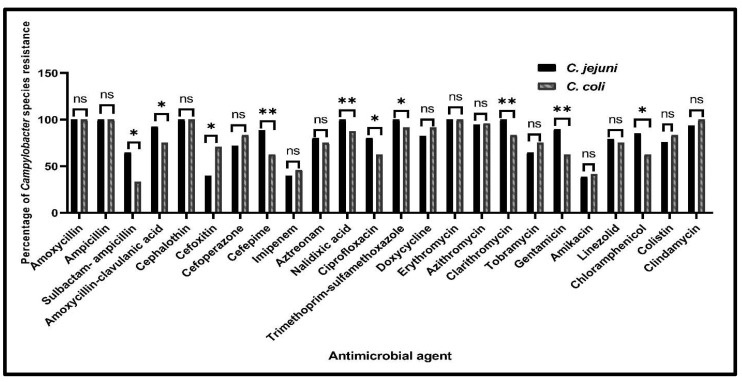
Prevalence of *Campylobacter jejuni* and *Campylobacter coli* resistance against 24 antimicrobial agents. ns: non-significant, * *p* < 0.05, ** *p* < 0.01.

**Figure 2 antibiotics-10-01342-f002:**
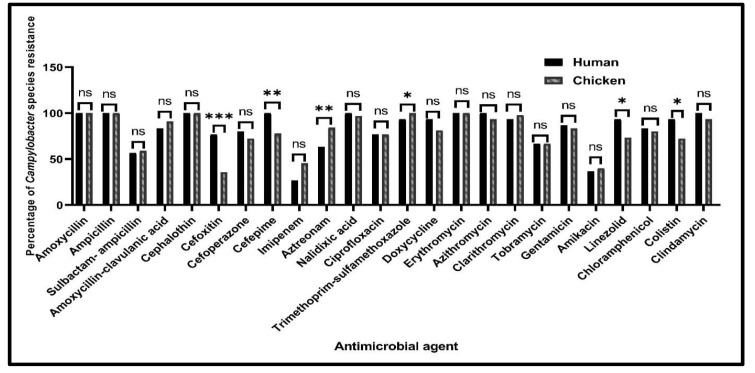
Prevalence of *Campylobacter* species resistance among human and chicken sources. ns: non-significant, * *p* < 0.05, ** *p* < 0.01, *** *p* < 0.001.

**Figure 3 antibiotics-10-01342-f003:**
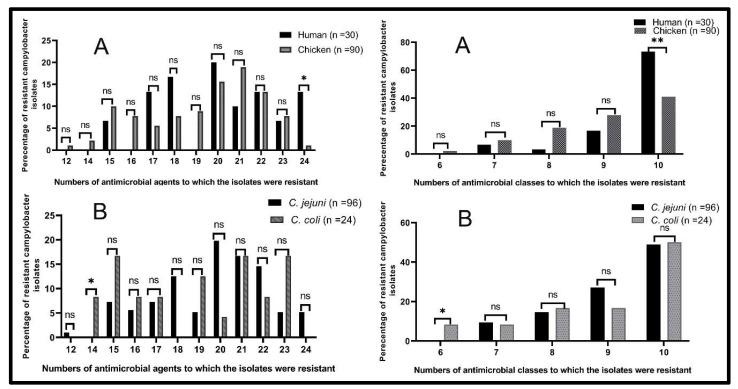
Antimicrobial resistance patterns of human and chicken *Campylobacter* isolates (**A**) and *C. jejuni* and *C. coli* (**B**). *n*: number, ns: non-significant, * *p* < 0.05, ** *p* < 0.01.

**Figure 4 antibiotics-10-01342-f004:**
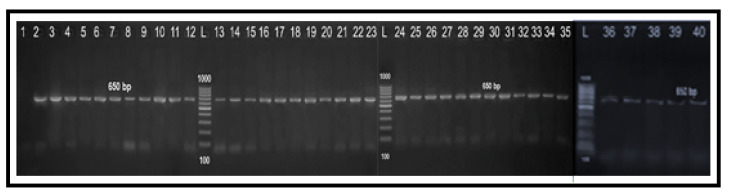
Agarose gel electrophoresis showing typical amplification products of *23S rRNA* gene for confirmation of genus *Campylobacter*. Lane L: 100 bp DNA ladder “Marker”, lane 1: negative control (PCR grade water), lane 2: positive control (*C. jejuni* NCTC11322), lanes 3–15: positive *Campylobacter* isolates from human stool swabs and lanes 16–40: positive *Campylobacter* isolates from chicken samples.

**Figure 5 antibiotics-10-01342-f005:**
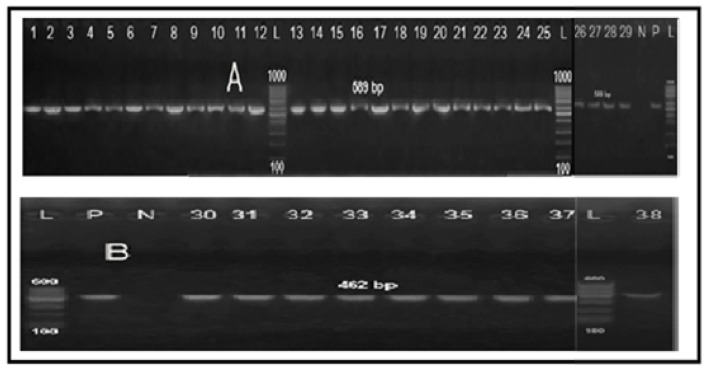
PCR amplification products of *mapA* gene specific for *C. jejuni* (**A**) and *ceuE* gene specific for *C. coli* (**B**). Lanes 1–11: *C. jejuni* from human origin, lanes 12–29: *C. jejuni* from chicken samples, lanes 30–31: *C. coli* from human samples, lane 32–38: *C. coli* from chicken origin, lane L: 100 bp DNA ladder “Marker”, lane P: positive controls (*C. jejuni* NCTC11322 and *C. coli* NCTC11366), lane N: negative control (PCR-grade water).

**Figure 6 antibiotics-10-01342-f006:**
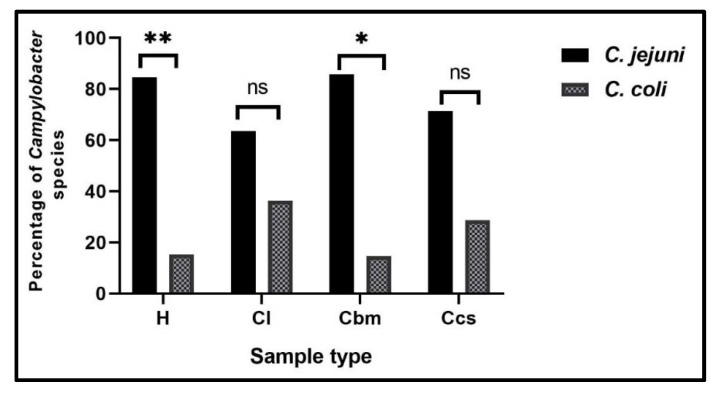
Prevalence of molecularly identified *Campylobacter* species in different samples. H: human stool swabs, Cl: chicken liver, Cbm: chicken breast meat, Ccs: chicken cloacal swabs, ns: non-significant, * *p*< 0.05, ** *p* < 0.01.

**Figure 7 antibiotics-10-01342-f007:**
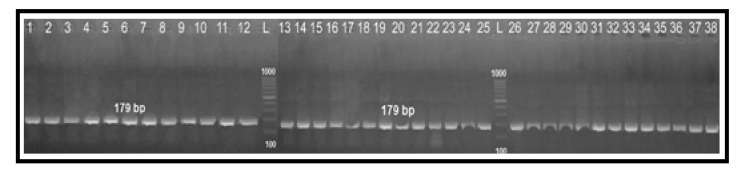
PCR-restriction fragment length polymorphism patterns obtained after digestion of *gyrA* PCR products with FastDigest *Rsa*I enzyme. Lane L: 100 bp DNA ladder and lanes 1–38: PCR-RFLP fragments obtained for *Campylobacter* species isolated from human and chicken samples.

**Table 1 antibiotics-10-01342-t001:** Prevalence of *Campylobacter* species in different samples at Sharkia Governorate, Egypt.

Sample Type (Symbol, No.)	Total No. of *Campylobacter* * Isolates (%)	No. of *Campylobacter* spp. (%) *	
		*C. jejuni*	*C. coli*
Human stool swabs (H, 35)	30 (85.7)	25 (71.4)	5 (14.3)
Broiler chicken samples (C, 175)	90 (51.4)	71 (40.6)	19 (10.9)
Cloacal swabs (Ccs, 35)	31 (88.6)	24 (68.6)	7 (20)
Breast muscles (Cbm, 35)	28 (80)	24 (68.6)	4 (11.4)
Liver (Cl, 35)	31 (88.6)	23 (65.7)	8 (22.9)
Chicken franks (Cf, 35)	0	0	0
Chicken luncheon meats (Cln, 35)	0	0	0
Total (210)	120 (57.1)	96 (45.7)	24 (11.4)

* The isolation rates were calculated concerning the total number of the examined samples.

**Table 2 antibiotics-10-01342-t002:** Antimicrobial resistance patterns of *Campylobacter* species isolated from different sources.

Antimicrobial Class	Antimicrobial Agent	No. of *C. jejuni* Isolates (%) (*n* = 96)		No. of *C. coli* Isolates (%) (*n* = 24)		Total No. of *Campylobacter* Isolates (%) (*n* = 120)
		Human (25)	Chicken (71)	Human (5)	Chicken (19)	
**Beta-lactams**	Amoxycillin	25 (100)	71 (100)	5 (100)	19 (100)	120 (100)
	Ampicillin	25 (100)	71 (100)	5 (100)	19 (100)	120 (100)
	Sulbactam-ampicillin	15 (60)	47 (66.2)	2 (40)	6 (31.6)	70 (58.3)
	Amoxycillin-clavulanic acid	23 (92)	66 (93)	2 (40)	16 (84.2)	107 (89.2)
	Cephalothin	25 (100)	71 (100)	5 (100)	19 (100)	120 (100)
	Cefoxitin	18 (72)	20 (28.2)	5 (100)	12 (63.2)	55 (45.8)
	Cefoperazone	19 (76)	50 (70.4)	5 (100)	15 (78.9)	89 (74.2)
	Cefepime	25 (100)	60 (84.5)	5 (100)	10 (52.6)	100 (83.3)
	Imipenem	6 (24)	32 (45.1)	2 (40)	9 (47.4)	49 (40.8)
	Aztreonam	14 (56)	63 (88.7)	5 (100)	13 (68.4)	95 (79.2)
**Quinolones**	Nalidixic acid	25 (100)	71 (100)	5 (100)	16 (84.2)	117 (97.5)
	Ciprofloxacin	21 (84)	56 (78.9)	2 (40)	13 (68.4)	92 (76.7)
**Sulfonamides**	Trimethoprim-sulfamethoxazole	25 (100)	71 (100)	3 (60)	19 (100)	118 (98.3)
**Tetracyclines**	Doxycycline	25 (100)	54 (76.1)	3 (60)	19 (100)	101 (84.2)
**Macrolides**	Erythromycin	25 (100)	71 (100)	5 (100)	19 (100)	120 (100)
	Azithromycin	25 (100)	66 (93)	5 (100)	18 (94.7)	114 (95)
	Clarithromycin	25 (100)	71 (100)	3 (60)	17 (89.5)	116 (96.7)
**Aminoglycosides**	Tobramycin	15 (60)	47 (66.2)	5 (100)	13 (68.4)	80 (66.7)
	Gentamicin	23 (92)	63 (88.7)	3 (60)	12 (63.2)	101 (84.2)
	Amikacin	8 (32)	29 (40.8)	3 (60)	7 (36.8)	47 (39.2)
**Oxazolidones**	Linezolid	25 (100)	51 (71.8)	3 (60)	15 (78.9)	94 (78.3)
**Phenicols**	Chloramphenicol	23 (92)	59 (83.1)	2 (40)	13 (68.4)	97 (80.8)
**Polypeptides**	Colistin	23 (92)	50 (70.4)	5 (100)	15 (78.9)	93 (77.5)
**Lincosamide**	Clindamycin	25 (100)	65 (91.5)	5 (100)	19 (100)	114 (95)

*n*: number.

**Table 3 antibiotics-10-01342-t003:** Multiple antibiotic resistance indices of *Campylobacter* species isolated from different sources.

MAR Index	No. of Antimicrobials to Which the Isolates Were Resistant	No. of AMC	No. of Resistant *Campylobacter* Isolates from Different Sources (%)				Total (120)	Character of Resistant Strains
			Human (30)		Chicken (90)			
			*C. jejuni* (25)	*C. coli* (5)	*C. jejuni* (71)	*C. coli* (19)		
0.50	12	7	-	-	1 (1.4)	-	1 (0.8)	MDR
0.58	14	6	-	-	-	2 (10.5)	2 (1.7)	
0.63	15	7	-	2 (40)	5 (7.1)	-	7 (5.8)	
		8	-	-	2 (2.8)	2 (10.5)	4 (3.3)	
0.67	16	8	-	-	5 (7)	-	5 (4.2)	
		10	-	-	-	2 (10.5)	2 (1.7)	
0.71	17	7	-	-	3 (4.2)	-	3 (2.5)	
		8	-	1 (20)	-	-	1 (0.8)	
		9	-	-	1 (1.4)	1 (5.3)	2 (1.7)	
		10	3 (12)	-	-	-	3 (2.5)	
0.75	18	8	-	-	2 (2.8)	-	2 (1.7)	
		9	2 (8)	-	3 (4.2)	-	5 (4.2)	
		10	3 (12)	-	2 (2.8)	-	5 (4.2)	
0.79	19	8	-	-	1 (1.4)	1 (5.3)	2 (1.7)	
		9	-	-	1 (1.4)	1 (5.3)	2 (1.7)	
		10	-	-	3 (4.2)	1 (5.3)	4 (3.3)	
0.83	20	9	2 (8)	-	4 (5.6)	-	6 (5)	
		10	4 (16)	-	9 (12.7)	1 (5.3)	14 (11.7)	
0.88	21	9	-	-	13 (18.3)	1 (5.3)	14 (11.7)	
		10	3 (12)	-	-	3 (15.8)	6 (5)	
0.92	22	8	-	-	4 (5.6)	-	4 (3.3)	XDR
		10	4 (16)	-	6 (8.5)	2 (10.5)	12 (10)	
0.96	23	9	-	1 (20)	-	-	1 (0.8)	
		10	-	1 (20)	5 (7)	2 (10.5)	8 (6.7)	
1	24	10	4 (16)	-	1 (1.4)	-	5 (4.2)	PDR

MAR: multiple antibiotic resistance, AMC: antimicrobial classes, MDR: multidrug resistant, XDR: extensively drug resistant, PDR: pan drug resistant.

**Table 4 antibiotics-10-01342-t004:** Minimal inhibitory concentrations of ciprofloxacin against *Campylobacter* isolates from different sources.

Isolates Source	*Campylobacter* Species	No. of *Campylobacter* Isolates Showing MIC Values of Ciprofloxacin (µg/mL) *							MIC_50_	MIC_90_
		4	8	16	32	64	128	≥256		
Chicken (25)	*C. jejuni* (18)	2	5	5	1	2	2	1	16	128
	*C. coli* (7)	2	3	2	-	-	-	-	8	16
Human (13)	*C. jejuni* (11)	2	2	3	-	-	-	4	16	256
	*C. coli* (2)	-	-	1	-	1	-	-	16	64
Total (38)		6	10	11	1	3	2	5	16	256

***** All isolates were resistant to ciprofloxacin (MIC ≥ 4 µg/mL), MIC: minimum inhibitory concentration, MIC_50_ = (*n* × 0.5), MIC_90_ = (*n* × 0.9).

**Table 5 antibiotics-10-01342-t005:** Sequences of oligonucleotide primers targeting four genes of *Campylobacter* species and their PCR-amplified products.

Specificity (Target Gene)	Primer Sequence (5′–3′)	Amplified Product (bp)	Reference
Genus *Campylobacter (23S rRNA)*	F: TATACCGGTAAGGAGTGCTGGAG	650	[49]
	R: ATCAATTAACCTTCGAGCACCG		
*Campylobacter coli (ceuE)*	F: AATTGA AAATTG CTCCAACTATG	462	[50]
	R: TGATTT TATTATTTGTAGCAGCG		
*Campylobacter jejuni (mapA*)	F: CTATTTTATTTTTGAGTGCTTGTG	589	[50]
	R: GCTTTATTTGCCATTTGTTTTATTA		
PCR-RFLP (*gyrA*)	F: AAATCAGCCCTATAGTGGGTGCTGTTATAGGTCGTTAT C ACCCACACATGGAGGT	179	[17]
	R: TCAGTATAACGCATCGCAGC		

## Data Availability

All data generated or analyzed during this study are included in the submitted manuscript.

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
