# Peer review of "Molecular Detection of Fluoroquinolone Resistance among Multidrug-, Extensively Drug-, and Pan-Drug-Resistant Campylobacter Species in Egypt"

_antibiotics, 2021, doi:10.3390/antibiotics10111342_

Round 1

Reviewer 1 Report

The manuscript by Ammar et al. is undoubtedly original, well designed and conducted, with original and relevant contribution to the understanding of the complex puzzle of the antimicrobial resistance of Campylobacter species in the Zagazig city, Sharkia Governorate, Egypt. A

I support its publication after appropriate major modifications as outlined below.

Title

Being a study conducted at regional level and in agreement with the presented data in the conclusion section the “Egypt” word need to be inserted

Introduction

Lines 40 – 43: to ensure the validity of statement please consult the yearly published EFSA report, referring to the etiology of human campylobacteriosis throughout the Europe

Line 47: the authors need to highlight the importance of the frequently occurred cross contaminations wit Campylobacter spp. within the household level in the disease maintenance and spreading

Line 41: the reference no. 8 is not appropriate

Lines 78-79: “little” or “no”? If “little” please add reference(s)

Materials and methods

Line 273: “chicken 272 (n=175) and human (n=35)” istead of „chicken 272 (175) and human (35)” and throughout the manuscript where you refer to numbers

Lines 271-280: the sampling collection strategy need to be detailed. How do you select the sampling interval, every week, month…? How many retail outlets, how many samples from each retail outlet? Number of samples every day? Does the Retail stores is always the same?

Line 283: I wonder, the authors followed exactly the International Standards Organization (ISO) guideline? Without slight modifications?

Lines 298-299: which was the strategy of the antimicrobial’s selection? They are regularly used in human and veterinary medicine in the screened area?

Line 308: “few” – uncler? Please be more explicit

Line 314: please change “sensitivity” with “susceptibility”

Line 316: I wonder the results were interpreted as Resistant, Intermediate or Susceptible? Please refer to this

Line 327: “Mueller-Hinton broth” instead of “Muller-Hinton broth”

Results

Line 99: statistically significant differences (p ˂ 0.001).

Line 105: “(100% each)” – please delete

Line 110: in the Table 2 „campylobacter” with sentence case

Line 110-111: insert a subheading of the Table indicating that “n”=number

Line 119: to represent the magnitude of significance within the whole manuscript please insert the recorded exact value for “p”

Lines 133, 152, 154, 159, 163, 173, 186, 223, 237, 241, 244, : Campylobacter spp. - italicized

Line 133-144: “were re-143 sistant to 7 (40%) and 8, 9 and 10 (20% each) antimicrobial classes.” – unclear, please rephrase

Lines 185 and 203: Considering that the molecular investigations are one of the most valuable part of the study, the authors need to find the way to insert representative electrophoresis gel images with the PCR results in the manuscript, to be more informative for the reader.

Discussion

Line 208, 209: “spp.” – not italicized, please be more consistent throughout the manuscript

Line 215: “previous studies carried out in Egypt; 32.8% [10], 27.3% [25] and 7.6% [26].” – it is contradictive to the statement retrieved within lines

Line 253-254: please insert a reference after the sentence

Conclusions

Line 364: the authors need to complete this section with the idea that the results of the present study offer useful insight to public health specialist in management of Campylobacter infections

Line 366: “isolate” – please uniformize the font

Line 369: “to reduce the occurrence of resistant” instead of “to reduce the resistant”

Finally, the authors need to insert a complex sentence highlighting the study limitation and further perspectives in the approached research area

Author Response

  1. The manuscript by Ammar et al. is undoubtedly original, well designed and conducted, with original and relevant contribution to the understanding of the complex puzzle of the antimicrobial resistance of Campylobacter species in the Zagazig city, Sharkia Governorate, Egypt.

Thank you for your positive comment on our manuscript.

I support its publication after appropriate major modifications as outlined below.

  1. Title : Being a study conducted at regional level and in agreement with the presented data in the conclusion section the “Egypt” word need to be inserted

Author response: thank you for your comment. We agree with your suggestions; thus, it was corrected in the revised manuscript at line 4

Introduction

  1. Lines 40 – 43: to ensure the validity of statement please consult the yearly published EFSA report, referring to the etiology of human campylobacteriosis throughout the Europe

Author response: thank you for your comment. We agree with your suggestions; thus, it was illustrated in the revised manuscript at lines 47-49, 473-480.

  1. Line 47: the authors need to highlight the importance of the frequently occurred cross contaminations with Campylobacter spp. within the household level in the disease maintenance and spreading

Author response: thank you for your comment. We agree with your suggestions; thus, it was illustrated in the revised manuscript at line 54-63.

  1. Line 41: the reference no. 8 is not appropriate

Author response: thank you for your comment. We agree with your suggestions; thus, it was corrected in the revised manuscript at line 491 and replaced with another reference (Ma et al., 2017).

  1. Lines 78-79: “little” or “no”? If “little” please add reference(s)

Author response: thank you for your comment. We agree with your suggestions; thus, it was illustrated in the revised manuscript at line 105.

Materials and methods

  1. Line 273: “chicken 272 (n=175) and human (n=35)” istead of „chicken 272 (175) and human (35)” and throughout the manuscript where you refer to numbers

Author response: thank you for your comment. We agree with your suggestions; thus, it was corrected in the revised manuscript at lines 336,339.

  1. Lines 271-280: the sampling collection strategy need to be detailed. How do you select the sampling interval, every week, month…? How many retail outlets, how many samples from each retail outlet? Number of samples every day? Does the Retail stores is always the same?

Author response: thank you for your comment. We agree with your suggestions; thus, it was illustrated in the revised manuscript at lines 336-339.

  1. Line 283: I wonder, the authors followed exactly the International Standards Organization (ISO) guideline? Without slight modifications?

Author response: thank you for your comment. Yes, we followed exactly the International Standards Organization (ISO) guideline as illustrated in the revised manuscript at lines 347-355.

  1. Lines 298-299: which was the strategy of the antimicrobial’s selection? They are regularly used in human and veterinary medicine in the screened area?

Author response: thank you for your comment. Yes, we chose the antimicrobial agents that were regularly used in human and veterinary medicine in Egypt; thus, it was illustrated in the revised manuscript at line 364.

  1. Line 308: “few” – uncler? Please be more explicit

Author response: thank you for your comment. We agree with your suggestions; thus, it was illustrated in the revised manuscript at line 373.

  1. Line 314: please change “sensitivity” with “susceptibility”

Author response: thank you for your comment. We agree with your suggestions; thus, it was corrected in the revised manuscript at line 379.

  1. Line 316: I wonder the results were interpreted as Resistant, Intermediate or Susceptible? Please refer to this

Author response: thank you for your comment. We agree with your suggestions; thus, it was illustrated in the revised manuscript at line 382.

  1. Line 327: “Mueller-Hinton broth” instead of “Muller-Hinton broth”

Author response: thank you for your comment. We agree with your suggestions; thus, it was corrected in the revised manuscript at line 394.

Results

  1. Line 99: statistically significant differences (p ˂ 0.001).

Author response: thank you for your comment. We agree with your suggestions; thus, it was corrected in the revised manuscript at line 126.

  1. Line 105: “(100% each)” – please delete

Author response: thank you for your comment. We agree with your suggestions; thus, it was corrected in the revised manuscript at line 132.

  1. Line 110: in the Table 2 „campylobacter” with sentence case

Author response: thank you for your comment. We agree with your suggestions; thus, it was corrected in the revised manuscript in the table cell.

  1. Line 110-111: insert a subheading of the Table indicating that “n”=number

Author response: thank you for your comment. We agree with your suggestions; thus, it was illustrated in the revised manuscript at line 139.

  1. Line 119: to represent the magnitude of significance within the whole manuscript please insert the recorded exact value for “p”

Author response: thank you for your comment. We agree with your suggestions; thus, it was illustrated in the revised manuscript at lines 150, 151, 163, 166, 195, 198.

  1. Lines 133, 152, 154, 159, 163, 173, 186, 223, 237, 241, 244, : Campylobacter spp. - italicized

Author response: thank you for your comment. We agree with your suggestions; thus, it was corrected throughout the revised manuscript.

  1. Line 133-144: “were re-143 sistant to 7 (40%) and 8, 9 and 10 (20% each) antimicrobial classes.” – unclear, please rephrase

Author response: thank you for your comment. We agree with your suggestions; thus, it was rephrased in the revised manuscript at line 175.

  1. Lines 185 and 203: Considering that the molecular investigations are one of the most valuable part of the study, the authors need to find the way to insert representative electrophoresis gel images with the PCR results in the manuscript, to be more informative for the reader.

Author response: thank you for your comment. We agree with your suggestions; thus, it was illustrated in the revised manuscript at lines 234-245 and 256-260.

Discussion

  1. Line 208, 209: “spp.” – not italicized, please be more consistent throughout the manuscript

Author response: thank you for your comment. We agree with your suggestions; thus, it was rephrased in the revised manuscript at lines 263.264.

  1. Line 215: “previous studies carried out in Egypt; 32.8% [10], 27.3% [25] and 7.6% [26].” – it is contradictive to the statement retrieved within lines

Author response: thank you for your comment. We agree with your suggestions; thus, it was rephrased in the revised manuscript at lines 268-270.

  1. Line 253-254: please insert a reference after the sentence

Author response: thank you for your comment. We agree with your suggestions; thus, it was corrected in the revised manuscript at line 315.

Conclusions

  1. Line 364: the authors need to complete this section with the idea that the results of the present study offer useful insight to public health specialist in management of Campylobacter infections

Author response: thank you for your comment. We agree with your suggestions; thus, it was corrected in the revised manuscript at line 438-443.

  1. Line 366: “isolate” – please uniformize the font

Author response: thank you for your comment. We agree with your suggestions; thus, it was corrected in the revised manuscript at line 433.

  1. Line 369: “to reduce the occurrence of resistant” instead of “to reduce the resistant”

Author response: thank you for your comment. We agree with your suggestions; thus, it was corrected in the revised manuscript at line 436.

  1. Finally, the authors need to insert a complex sentence highlighting the study limitation and further perspectives in the approached research area

Author response: thank you for your comment. We agree with your suggestions; thus, it was corrected in the revised manuscript at lines 444-445.

Reviewer 2 Report

Nice work

Line 51- persons should be replaced by patients

Line 67- Could there be any other mechanisms responsible for fluroquinolone resistance?  If given isolates are multidrug resistant then potentially more than one mechanisms would be in action. In this case they could be mutations in genes involved in drug transportation, drug metabolism.  Please elaborate on such plausible causes.

Line 76- Kindly provide molecular biology in brief manner behind PCR-RFLP.  Are there any other methods that could be used for the mutation detection.

Line 206- As mutation Thr-86 is detected in all 38 campy. isolates and respective phenotype is seen in the form of ciprofloxacin MIC values, potential role of Thr-86 has to be discussed in discussion section.  What type of mutation is this and which amino acid/ protein is affected by this mutation.  Is this mutation seen only on chromosomal gene or is it seen on genes located on mobile genetic elements like plasmids.  If later is the case, then can explain horizontal spread of the resistance.    

Author Response

Reviewer Comments:

  1. Nice work

Thank you for your comment

  1. Line 51- persons should be replaced by patients

Author response: thank you for your comment. It was done and corrected in the revised manuscript at line 67.

  1. Line 67- Could there be any other mechanisms responsible for fluroquinolone resistance? If given isolates are multidrug resistant then potentially more than one mechanisms would be in action. In this case they could be mutations in genes involved in drug transportation, drug metabolism.  Please elaborate on such plausible causes.

Author response: thank you for your comment. We agree with your suggestions; thus, it was illustrated in the revised manuscript at lines 81-87.

  1. Line 76- Kindly provide molecular biology in brief manner behind PCR-RFLP.

Author response: thank you for your comment. We agree with your suggestions; thus, it was illustrated in the revised manuscript at lines 100-102.

  1. Are there any other methods that could be used for the mutation detection.

Author response: Yes, there was other methods that could be used for the mutation detection and it was illustrated in the revised manuscript at lines 92-95.

  1. Line 206- As mutation Thr-86 is detected in all 38 campy. isolates and respective phenotype is seen in the form of ciprofloxacin MIC values, potential role of Thr-86 has to be discussed in discussion section. What type of mutation is this and which amino acid/ protein is affected by this mutation.  Is this mutation seen only on chromosomal gene or is it seen on genes located on mobile genetic elements like plasmids.  If later is the case, then can explain horizontal spread of the resistance.  

Author response: thank you for your comment. We agree with your suggestions; thus, it was illustrated in the revised manuscript at lines 252-254 (the result section) and lines 320-322 (the discussion section).

Round 2

Reviewer 1 Report

The authors correctly acknowledged all of the raised concerns.

Congratulation!